# COVID-19 Vaccine Acceptance and Beliefs among Black and Hispanic Americans

**Katherine Kricorian**  *, Karin Turner

MiOra, Encino, California, United States of America

* krn830@gmail.com

**Data Availability Statement:** All relevant data are within the paper and its Supporting Information files.

**Funding:** The authors received no specific funding for this work.

## Abstract

The introduction of COVID-19 vaccines is a major public health breakthrough. However, members of US Black and Hispanic communities, already disproportionately affected by the COVID-19 virus, may be less willing to receive the vaccine. We conducted a broad, representative survey of US adults (N = 1,950) in order to better understand vaccine beliefs and explore opportunities to increase vaccine acceptance among these groups. The survey results suggested that Black and Hispanic individuals were less willing than Whites to receive the vaccine. US Blacks and Hispanics also planned to delay receiving the COVID-19 vaccine for a longer time period than Whites, potentially further increasing the risk of contracting COVID-19 within populations that are already experiencing high disease prevalence. Black respondents were less likely to want the COVID-19 vaccine at all compared with Whites and Hispanics, and mistrust of the vaccine among Black respondents was significantly higher than other racial/ethnic groups. Encouragingly, many Black and Hispanic respondents reported that COVID-19 vaccine endorsements from same-race medical professionals would increase their willingness to receive it. These respondents said they would also be motivated by receiving more information on the experiences of vaccine study participants who are of their own race and ethnicity. The results have implications for improved messaging of culturally-tailored communications to help reduce COVID-19 vaccine hesitancy among communities disproportionately impacted by the pandemic.

## Introduction

The evolving COVID-19 pandemic is one of the greatest public health threats to emerge in modern times and has disproportionately impacted communities of color in the United States. Black and Hispanic Americans have experienced particularly high rates of morbidity and mortality [1, 2]. At the time of this writing, approximately fifty thousand Black Americans and seventy thousand Hispanic Americans have died from the virus. Black and Hispanic Americans are also more likely to suffer from pre-existing health conditions such as diabetes, obesity, and coronary artery disease, which exacerbate the effects of COVID-19 infection [3]. The unprecedented impact of the COVID-19 pandemic on these groups is likely also contributed to by a variety of social factors that have increased disease transmission within minority communities and reduced their ability to receive adequate healthcare [3]. Further compounding these issues,

**Competing interests:** The authors have declared no competing interests exist.

the COVID-19 pandemic has acutely impacted Black and Hispanic Americans economically. Black and Hispanic workers have experienced pandemic-related layoffs and job losses to a greater degree than those in many other racial and ethnic groups [4]. Research also indicates that Hispanics in particular are postponing health services, which may only make COVID-19's longer-term impact worse [5].

Recently, COVID-19 vaccines have become broadly available in the US. However, some Blacks and Hispanics may be reluctant to receive a vaccination [6]. Unethical medical incidents such as the Tuskegee experiments in 1932–1972, where Black men infected with syphilis were purposely denied treatment and eugenics laws that were disproportionately applied to people of color have left a long-lasting impact on ethnic minority populations, often diminishing their trust in the healthcare establishment [7, 8]. This lingering distrust may contribute to vaccine hesitancy. Improved understanding of COVID-19 perceptions among Black and Hispanic Americans is needed to slow the infection rate and achieve herd immunity. Anderson et al. [9] noted that the vaccination levels required to achieve COVID-19 herd immunity may be up to 90%, making it necessary to achieve near-universal levels of vaccine acceptance. However, Dong et al. [10] hypothesized that vaccination of 60% of the population may be sufficient. Randolph and Barriero [11] noted that herd immunity thresholds can vary widely within populations and are non-uniformly distributed, implying that a single standard for herd immunity may be an oversimplification.

Vaccine mandates, such as those required for school entry or public benefits, are common in many countries and have experienced wide public support, especially in the wake of disease outbreaks such as the 2015 measles outbreak in California and the 2017 measles outbreak in Italy [12]. In their review, Spencer, Trondsen-Pawlowski and Thomas [13] reported the incidence of serious adverse advents from vaccinations is generally very small, especially when considered in light of the vaccines' disease prevention benefits. Still, some studies have found that people of color may be more reluctant than Whites to vaccinate themselves and their children against diseases such as influenza. Black parents and patients have been shown to have higher concern about factors such as vaccine-related side effects [14]. These concerns resulted in lower influenza vaccination rates: during the 2019–2020 flu season, the Centers for Disease Control (CDC) estimated that just 38% of Hispanic individuals and 41% of Black individuals received an influenza vaccine, compared to over half of non-Hispanic Whites. CDC data also indicate that Black and Hispanic Americans were especially likely to contract the flu, possibly due to vaccine reluctance compounded by the health impact of other social and economic inequities [15]. The World Health Organization (WHO) has identified vaccine hesitancy as one of the top ten threats to global health, stating that vaccine hesitancy is a major factor driving the resurgence of measles [16]. Notably, the spread of misinformation, often through the media, may be a contributor to vaccine hesitancy in a variety of groups [17–19].

In the era of COVID-19, vaccine hesitancy may have even greater consequences. Novel and dangerous SARS-CoV-2 variants continue to emerge, making the pursuit of herd immunity through wide-scale, global vaccination an urgent imperative. Breaking the chain of disease transmission for SARS-CoV-2 is critical. Anderson et al. [20] discussed the impact of child vaccination programs and observed that beyond the direct effect on the children who are vaccinated, child vaccination also provides substantial indirect community protection. By blocking transmission, vaccinated people of all ages prevent exposure to unvaccinated members of their communities. Anderson et al. [20] further noted that when new childhood vaccines are implemented, disease incidence significantly declines throughout the entire population, even among those who are unable to receive the vaccines themselves.

Although vaccines in general have been shown to greatly reduce disease mortality rates safely and effectively, many people of all racial and ethnic backgrounds still fear that vaccines are unsafe [16, 21]. Harrison and Wu [21] noted that a lack of understandable, scientifically

accurate vaccine information may be another reason for vaccine hesitancy, along with racial/ethnic underrepresentation among authorities leading vaccine programs. For groups such as Blacks and Hispanics, perceptions of not being adequately represented by the healthcare experts forming, regulating, and distributing the vaccine may be a particular issue that discourages them from receiving it.

This study aims to further explore COVID-19 vaccine hesitancy and examine factors that may help remedy such hesitancy among Black and Hispanic Americans. A better understanding of COVID-19 vaccine hesitancy among people of color can also contribute to customized health communication strategies within minority communities and help increase future vaccine uptake beyond COVID-19.

## Hypotheses

The first hypothesis (H1) was that Black and Hispanic respondents will be less willing to receive the COVID-19 vaccine than Whites. The second hypothesis (H2) was that Black and Hispanic respondents will plan to wait longer to receive the COVID-19 vaccine than Whites. The third hypothesis (H3) was that Black and Hispanic respondents will have higher levels of mistrust about the COVID-19 vaccine than White respondents.

## Methods

An online survey was conducted in January 2021. The survey assessed demographics, COVID-19 vaccine experiences, and COVID-19 and vaccine beliefs of respondents. The first section of the survey asked respondents to indicate their COVID-19 perceptions, experiences, and knowledge. The second part of the survey analyzed COVID-19 vaccine awareness and opinions. The third and final part of the survey included questions about demographics and experiences with healthcare. The survey consisted of questions that were adapted from previously validated surveys and was conducted in English [22, 23]. Respondents were recruited through an E-mail invitation from an online survey panel and response was voluntary. The survey response rate to the email invitation was 9%. This response rate is in line with those of other, large-scale online surveys.

Prior to data collection, the research protocol was reviewed by WCG Institutional Review Board and classified as exempt under 45 CFR § 46.104(d)(2), because the research was an anonymous survey study. Consent to participate was obtained after respondents were given sufficient information about the survey and asked if they wanted to complete an anonymous survey. In order to protect subject privacy, no names or identifiers were collected from survey respondents. Respondents were recruited from across the United States. A broad, national sample of US respondents was collected with demographics (age, gender, race/ethnicity and household income) approximating percentages from the United States Census, and data was further weighted to represent the US population more precisely. Population values were developed using data from the 2019 US Census American Community Survey [24]. Population percentages were calculated for the matrixed percentage values for age range, gender, race/ethnicity, household income range and Census region, and were compared to the same matrixed percentage values within the survey sample. Survey responses within each Age range * Gender * Race/ethnicity * income range * Census region cell were upweighted or downweighted so that the resulting percentages, shown in Table 1, reflected the population distribution present in the American Community Survey results. Most cell weights differed negligibly from 1.0. Respondents gave their consent prior to starting the survey. Prior to the beginning of the survey, respondents were asked to indicate their age. Prospective respondents under age 18

**Table 1. Survey sample characteristics.**

| | | Overall | White | Black | Hispanic | Asian | Other |
|---|---|---|---|---|---|---|---|
| | N | 1950 | 1248 | 235 | 304 | 32 | 130 |
| **Place of Residence** | **Urban** | 32.2% | 25.4% | 49.8% | 46.1% | 42.4% | 30.5% |
| | **Suburban** | 45.1% | 47.7% | 36.2% | 40.8% | 51.5% | 45.0% |
| | **Rural** | 22.6% | 26.7% | 14.0% | 13.2% | 6.1% | 24.4% |
| **Residence Type** | **Detached house** | 57.3% | 62.4% | 46.2% | 52.3% | 56.3% | 40.0% |
| | **Attached house or townhouse** | 10.0% | 8.4% | 11.5% | 11.2% | 9.4% | 19.2% |
| | **Apartment or flat** | 22.7% | 18.3% | 36.8% | 27.3% | 31.3% | 26.9% |
| | **Other** | 10.1% | 10.9% | 5.6% | 9.2% | 3.1% | 13.7% |
| **Living Companions** | **Alone** | 19.8% | 19.6% | 24.7% | 15.8% | 19.4% | 22.3% |
| | **With Adults (Only)** | 49.6% | 51.7% | 45.5% | 43.8% | 44.8% | 51.5% |
| | **With Children** | 30.6% | 28.7% | 29.8% | 40.5% | 35.8% | 26.2% |
| **Marital Status** | **Married or Living with Partner** | 53.1% | 57.1% | 40.3% | 50.7% | 50.0% | 43.8% |
| | **Divorced, Widowed or Separated** | 17.9% | 20.9% | 14.4% | 11.5% | 9.4% | 13.1% |
| | **Not Married or Living with Partner** | 29.0% | 22.0% | 45.3% | 37.8% | 40.6% | 43.1% |
| **2019 Annual HH Income** | **Median** | $51.6K | $54.1K | $37.5K | $46.4K | $67.4K | $35.3K |
| | **Less than $35K** | 36.5% | 32.8% | 47.6% | 39.1% | 25.8% | 50.0% |
| | **$35K to $74.9K** | 32.6% | 33.2% | 32.2% | 32.0% | 29.0% | 30.2% |
| | **$75K or more** | 30.9% | 34.0% | 20.3% | 29.0% | 45.2% | 19.8% |
| **2020 Annual HH Income** | **Median** | $48.0K | $51.9K | $37.9K | $44.9K | $65.5K | $32.5K |
| | **Less than $35K** | 38.6% | 35.3% | 47.1% | 41.4% | 26.7% | 52.6% |
| | **$35K to $74.9K** | 31.8% | 31.9% | 33.3% | 31.2% | 33.3% | 28.9% |
| | **$75K or more** | 29.6% | 32.9% | 19.6% | 27.5% | 40.0% | 18.4% |
| **Age** | **Median** | 46.6 yrs | 49.7 yrs | 44.6 yrs | 39.0 yrs | 44.8 yrs | 39.6 yrs |
| **Education Level** | **High School or Less** | 26.6% | 27.6% | 24.7% | 25.7% | 12.5% | 25.9% |
| | **Some Post-Secondary** | 38.6% | 36.4% | 48.1% | 36.9% | 25.1% | 48.9% |
| | **College Graduate or More** | 34.7% | 35.7% | 26.8% | 37.5% | 62.5% | 25.2% |

Notes: Hispanics may be any race, and members of other groups indicated they were non-Hispanic.

years old (N = 44) were thanked for their participation and terminated from the study. A sample of N = 1,950 adults aged 18 years and older completed the survey.

Data were analyzed with IBM SPSS Statistics 27 using $\chi^2$ tests and z-tests to compare specific groups to each other. Survey results from all five racial/ethnic groups analyzed are shown in the Tables. Analysis focused primarily on the results from the largest three of these five groups: White, Black and Hispanic, given the small sample size for the Asian group and the difficulty in meaningfully interpreting results from the Other category, which was comprised of a variety of different racial and ethnic groups, such as such as American Indian, Middle Eastern and multiracial individuals.

In the first portion of the survey, respondents were asked to indicate their attitudes regarding COVID-19. Then survey respondents were questioned regarding their awareness and assumptions about the COVID-19 vaccine. In the final section of the survey, respondents answered questions regarding their demographics. At the end of the survey, respondents were thanked for their participation.

## Results

Overall, the mean age of respondents was 46.6 years (SD = 17.4 years). The sample size for non-Hispanic Whites was N = 1,248 and the mean age was 49.7 years (SD = 17.3 years). Non-

Hispanic Blacks had a survey sample N = 235, with a mean age of 44.5 years (SD = 16.6 years). Hispanics of any race accounted for N = 304 respondents with a mean age of 39.0 years (SD = 14.8 years). The survey dataset also included N = 32 Asians (mean age = 44.8 years, SD = 16.3 years) and N = 130 respondents who classified themselves as "Other," or another race/ethnicity (mean age = 39.6 years, SD = 14.6 years). Approximately half (49.3%) of survey respondents identified as male and the remaining 50.7% identified as female. No respondents identified their gender as non-binary, although this option was offered as a response to the gender identification question. Overall self-reported median annual household income in 2019 was $51.6K, and in 2020 was lower, at $48.0K. People self-identifying as Asian reported the highest median incomes in both 2019 and 2020 ($67.4K and $65.5K, respectively) and those in the Other category reported the lowest median incomes (2019 median = $35.3K and 2020 median = $32.5K, see Table 1). In terms of household composition, 30.5% overall reported living in a household with one or more children under age 18, 49.7% reported living only with other adults, and 19.8% reported living alone. When asked to describe the location of their homes, 32.2% said it was urban, 45.2% said it was suburban and 22.6% said it was rural. Non-Hispanic Whites were most likely to report living in a suburb (45.1%) or a rural area (26.7%). Black, Hispanic and Asian respondents were less likely to be rural or suburban than Whites, and more likely to live in an urban area (see Table 1).

When asked if they are willing to receive the COVID-19 vaccine, fewer than 40% of respondents overall indicated they would "definitely" be willing to receive it. More than 20% said they would "probably not" or "definitely not" receive it. Race/ethnicity was significantly associated with likelihood to receive the vaccine ($\chi^2$ = 104.32, df = 16, n = 1950, p < .05). As shown in Table 2, approximately half as many Black respondents as compared with Whites stated that they would definitely be willing to receive the vaccine (22.6% for Blacks vs. 43.6% for Whites, p<0.05). Similarly, about 20% of Blacks stated they would definitely not receive the vaccine compared with 12.7% of Whites (p<0.05). A meaningful proportion of Hispanics (14.8%) stated that they would definitely not be willing to receive the vaccine, although this was not significantly different from White respondents. **H1 was partially confirmed.**

**Table 2. Likelihood to get the COVID-19 vaccine.**

| | | Race/ Ethnicity | | | | | |
|---|---|---|---|---|---|---|---|
| | | White | Black | Hispanic | Asian | Other | Total |
| After the COVID-19 vaccine is broadly available, how long do you plan to wait to get the vaccine? | I want to get it immediately | 38.5%[a] | 20.1%[b] | 28.2%[c] | 35.5%[a, b, c] | 26.7%[b, c] | 33.9% |
| | Up to 1 month | 12.3%[a] | 6.0%[b] | 13.8%[a] | 16.1%[a] | 11.5%[a, b] | 11.7% |
| | 2–3 months | 10.7%[a, b] | 12.4%[a, b, c] | 17.7%[c] | 19.4%[b, c] | 6.9%[a] | 11.9% |
| | 4–6 months | 9.9%[a] | 11.5%[a] | 11.8%[a] | 12.9%[a] | 3.8%[b] | 10.1% |
| | 7–9 months | 2.5%[a] | 7.3%[b] | 0.7%[c] | 3.2%[a, b, c] | 0.8%[a, c] | 2.7% |
| | 10–12 months | 2.2%[a] | 5.1%[b] | | 3.0%[a, b] | 3.1%[a, b] | 2.7% |
| | More than one year | 5.1%[a] | 9.8%[b] | 8.2%[b] | 6.5%[a, b] | 12.2%[b] | 6.7% |
| | I do not plan to get the COVID-19 vaccine | 18.7%[a] | 27.8%[b] | 16.7%[a] | 6.5%[a] | 35.1%[b] | 20.4% |
| Total | | 100.0% | 100.0% | 100.0% | 100.0% | 100.0% | 100.0% |

Notes: Hispanics may be any race, and members of other groups indicated they were non-Hispanic.

Each subscript letter denotes a subset of race/ethnicity categories whose column proportions do not differ significantly from each other at the p < .05 level.

**Table 3. Intended wait time to get the COVID-19 vaccine, after vaccine is broadly available.**

| | | Race/ Ethnicity | | | | | |
| --- | --- | --- | --- | --- | --- | --- | --- |
| | | White | Black | Hispanic | Asian | Other | Total |
| **After the COVID-19 vaccine is broadly available, how long do you plan to wait to get the vaccine?** | **I want to get it immediately** | $38.5\%_a$ | $20.1\%_b$ | $28.2\%_c$ | $35.5\%_{a, b, c}$ | $26.7\%_{b, c}$ | 33.9% |
| | **Up to 1 month** | $12.3\%_a$ | $6.0\%_b$ | $13.8\%_a$ | $16.1\%_a$ | $11.5\%_{a, b}$ | 11.7% |
| | **2–3 months** | $10.7\%_{a, b}$ | $12.4\%_{a, b, c}$ | $17.7\%_c$ | $19.4\%_{b, c}$ | $6.9\%_a$ | 11.9% |
| | **4–6 months** | $9.9\%_a$ | $11.5\%_a$ | $11.8\%_a$ | $12.9\%_a$ | $3.8\%_b$ | 10.1% |
| | **7–9 months** | $2.5\%_a$ | $7.3\%_b$ | $0.7\%_c$ | $3.2\%_{a, b, c}$ | $0.8\%_{a, c}$ | 2.7% |
| | **10–12 months** | $2.2\%_a$ | $5.1\%_b$ | $3.0\%_{a, b}$ | 0% | $3.1\%_{a, b}$ | 2.7% |
| | **More than one year** | $5.1\%_a$ | $9.8\%_b$ | $8.2\%_b$ | $6.5\%_{a, b}$ | $12.2\%_b$ | 6.7% |
| | **I do not plan to get the COVID-19 vaccine** | $18.7\%_a$ | $27.8\%_b$ | $16.7\%_a$ | $6.5\%_a$ | $35.1\%_b$ | 20.4% |
| **Total** | | 100.0% | 100.0% | 100.0% | 100.0% | 100.0% | 100.0% |

Notes: Hispanics may be any race, and members of other groups indicated they were non-Hispanic.

Each subscript letter denotes a subset of race/ethnicity categories whose column proportions do not differ significantly from each other at the p < .05 level.

There was a significant relationship between when respondents planned to receive the vaccine and their race/ethnicity ($\chi^2$ = 129.84, df = 28, n = 1950, p < .05). When asked how long they planned to wait before receiving the COVID-19 vaccine, Blacks and Hispanics were significantly less likely than Whites to state that they wanted to get vaccinated immediately (20.1% for Blacks, 28.2% for Hispanics and 38.5% for Whites, both comparisons p<0.05, see Table 3). Black and Hispanic respondents were also significantly more likely than Whites to indicate that they planned to wait for over a year to get the vaccine (9.8% for Blacks, 8.2% for Hispanics and 5.1% for Whites, both comparisons p<0.05). **H2 was confirmed**.

Overall, about half of respondents (51.4%) felt the COVID-19 vaccine "is going to be effective" and 40.8% agreed it "is going to be safe" and both variables showed a significant relationship with race/ethnicity overall (respectively: $\chi^2$ = 29.58, df = 4, n = 1950, p < .05 and $\chi^2$ = 11.42, df = 4, n = 1950, p < .05; see Table 4). More than a third (35.1%) of respondents indicated that the COVID-19 vaccine "is being rushed out too quickly," 25.4% said the COVID-19 vaccine is "just not something I trust" and 24.4% felt the vaccine "is going to be distributed

**Table 4. Agreement with statements about the COVID-19 vaccine (% "yes").**

| | White | Black | Hispanic | Asian | Other | |
| --- | --- | --- | --- | --- | --- | --- |
| **Going to be effective.** | $55.1\%_a$ | $38.7\%_b$ | $49.3\%_{a, c}$ | $60.6\%_a$ | $40.8\%_{b, c}$ | 51.4% |
| **Going to be safe.** | $42.7\%_a$ | $33.6\%_b$ | $41.1\%_{a, b}$ | $50.0\%_{a, b}$ | $32.8\%_b$ | 40.8% |
| **Being rushed out too quickly.** | $30.1\%_a$ | $47.7\%_b$ | $42.1\%_{b, c}$ | $25.0\%_{a, c}$ | $46.9\%_b$ | 35.1% |
| **Just not something I trust.** | $23.7\%_a$ | $34.0\%_b$ | $27.0\%_{a, b, c}$ | $15.2\%_{a, c}$ | $31.5\%_{b, c}$ | 25.8% |
| **Going to be distributed fairly.** | $25.7\%_a$ | $25.1\%_a$ | $24.0\%_a$ | $25.0\%_{a, b}$ | $12.2\%_b$ | 24.4% |
| **Not worth the risk.** | $15.6\%_a$ | $18.3\%_{a, b}$ | $17.1\%_a$ | $9.1\%_a$ | $25.4\%_b$ | 16.7% |
| **Dangerous.** | $10.0\%_a$ | $16.2\%_b$ | $13.5\%_{a, b}$ | $6.3\%_{a, b}$ | $13.7\%_{a, b}$ | 11.5% |
| **More harmful than getting COVID-19.** | $7.8\%_a$ | $12.3\%_b$ | $8.6\%_{a, b}$ | $6.1\%_{a, b}$ | $11.5\%_{a, b}$ | 8.7% |
| **Going to cause people to catch COVID-19.** | $6.4\%_a$ | $14.0\%_{b, c}$ | $7.9\%_a$ | $3.1\%_{a, c}$ | $16.8\%_b$ | 8.2% |

Notes: Hispanics may be any race, and members of other groups indicated they were non-Hispanic.

Each subscript letter denotes a subset of race/ethnicity categories whose column proportions do not differ significantly from each other at the p < .05 level.

Percentages total more than 100% because survey respondents could select multiple answers.

fairly." All three variables overall differed significantly by race/ethnicity ($\chi^2 = 45.75$, $\chi^2 = 15.56$ and $\chi^2 = 11.77$, respectively; in all three cases, df = 4, n = 1950, p < .05). A meaningful percentage of respondents agreed with the statements that the COVID-19 vaccine is "not worth the risk" (16.7%), "dangerous" (11.5%), or even "more harmful than getting COVID-19" (8.7%). "Not worth the risk" and "dangerous" differed significantly overall by race, with $\chi^2 = 9.95$ and $\chi^2 = 10.44$, both df = 4, n = 1950, p < .05. The variable "more harmful than getting COVID-19" did not differ overall by race/ethnicity ($\chi^2 = 6.84$, df = 4, n = 1950, ns).

Blacks were significantly less likely than Whites to believe that the vaccine would be effective (38.7% vs. 55.1%, p<0.05), along with 49.3% of Hispanics. Among Black respondents, the percentage saying the COVID-19 vaccine is "just not something I trust" was significantly higher than among Whites (34.0% vs. 23.7%%, p<0.05; see Table 4). A substantial proportion of Hispanics also indicated that the vaccine is "just not something I trust" (27.0%). Black respondents were also significantly less likely than Whites to agree that "the COVID-19 vaccine is going to be safe" (33.6% for Blacks vs. 42.7% for Whites, p<0.05) and more likely to believe that the vaccine is dangerous (16.2% vs. 10.0%, p<0.05). Blacks were also more likely than Whites to believe that the vaccine is more harmful than COVID-19 (12.3% vs. 7.8%, p<0.05) and that the vaccine would cause them to catch COVID-19 (14.0% of Blacks vs. 6.4% of Whites, p<0.05). Blacks were also less likely than Whites to feel that the vaccine is not worth the risk (18.3% of Blacks vs. 15.6% of Whites, p<0.05). Blacks and Hispanics were both significantly more likely than Whites to believe that the vaccine is "being rushed out too quickly" (47.7% of Blacks vs. 42.1% of Hispanics and 30.1% of Whites, p<0.05 for both comparisons; see Table 4). **H3 was generally confirmed, as differences between Whites and Hispanics were directional and did not all reach conventional levels of statistical significance.**

As shown in Table 5, nearly half (48.1%) of respondents overall said they would recommend that friends and family get the COVID-19 vaccine, 9.4% would recommend not getting the vaccine, and 42.6% would not make a recommendation either way. Responses to this survey question varied significantly by race/ethnicity ($\chi^2 = 80.20$, df = 8, n = 1950, p < .05). Both Blacks and Hispanics were significantly more likely than Whites to recommend that friends or family members not get the vaccine (11.1% for Blacks, 10.9% for Hispanics and 7.1% for Whites; p<0.05 for both comparisons). Whites and Hispanics were similarly likely to say they would recommend friends and family get the COVID-19 vaccine (51.2% and 50.3%). Black respondents, however, were significantly less likely to recommend the vaccine (32.3%, comparisons vs. Whites and Hispanics both p < .05). Whites and Hispanics were similarly likely to say they would not make a recommendation to friends or family either way about getting the COVID-19 vaccine (41.6% vs. 38.8%, respectively, ns). Blacks were significantly more likely than both groups not to make a suggestion either way (56.6%, comparisons vs. Whites and Hispanics both p<0.05).

Responses to the question "How important is it to you that a medical professional of your race/ethnicity endorses the vaccine before you take it?" differed significantly by race/ethnicity ($\chi^2$

**Table 5. Likelihood to recommend COVID-19 vaccination to friends/family.**

| | | Race/ Ethnicity | | | | | |
| --- | --- | --- | --- | --- | --- | --- | --- |
| | | White | Black | Hispanic | Asian | Other | Total |
| **Would you be likely to...** | **Recommend to your friends/family that they get vaccinated.** | 51.2%a | 32.3%b | 50.3%a | 56.3%a, c | 38.5%b, c | 48.1% |
| | **Recommend to your friends/family that they NOT get vaccinated.** | 7.1%a | 11.1%b | 10.9%b | 3.1%a, b | 26.2%c | 9.4% |
| | **Not make a suggestion either way.** | 41.6%a | 56.6%b | 38.8%a | 40.6%a, b | 35.4%a | 42.6% |
| | | 100.0% | 100.0% | 100.0% | 100.0% | 100.0% | 100.0% |

Notes: Hispanics may be any race, and members of other groups indicated they were non-Hispanic.

Each subscript letter denotes a subset of race/ethnicity categories whose column proportions do not differ significantly from each other at the p < .05 level.

**Table 6. Importance of COVID-19 vaccine endorsement by same-race medical professional.**

| | | Race/Ethnicity | | | | | |
|---|---|---|---|---|---|---|---|
| | | White | Black | Hispanic | Asian | Other | Total |
| **How important is it to you that a medical professional of your race/ethnicity endorses the vaccine before you take it?** | **Extremely** | 7.3%$_a$ | 12.0%$_b$ | 11.8%$_b$ | 9.1%$_{a, b}$ | 3.1%$_a$ | 8.3% |
| | **Very** | 9.0%$_a$ | 28.2%$_b$ | 14.8%$_c$ | 12.1%$_{a, c}$ | 16.8%$_c$ | 12.8% |
| | **Somewhat** | 13.3%$_a$ | 24.4%$_b$ | 19.1%$_b$ | 33.3%$_b$ | 19.8%$_b$ | 16.3% |
| | **Not very** | 14.6%$_a$ | 11.5%$_a$ | 15.5%$_a$ | 21.2%$_a$ | 15.3%$_a$ | 14.5% |
| | **Not at all** | 55.8%$_a$ | 23.9%$_b$ | 38.8%$_{c, d}$ | 24.2%$_{b, d}$ | 45.0%$_c$ | 48.1% |
| | | 100.0% | 100.0% | 100.0% | 100.0% | 100.0% | 100.0% |

Notes: Hispanics may be any race, and members of other groups indicated they were non-Hispanic.

Each subscript letter denotes a subset of race/ethnicity categories whose column proportions do not differ significantly from each other at the p < .05 level.

= 154.70, df = 16, n = 1950, p < .05). Blacks and Hispanics were more likely than Whites to state that it was "extremely" important that a medical professional of their same race/ethnicity endorsed the vaccine before they took it (12.0% for Blacks vs. 11.8% for Hispanics, and 7.3% for Whites, both comparisons vs. Whites p<0.05; see Table 6). Blacks and Hispanics were significantly less likely than Whites to state that it is "not at all" important to them that a medical professional of their race/ethnicity endorses the vaccine before they take it (23.9% for Blacks vs. 38.8% for Hispanics and 55.5% for Whites, p<0.05 for both comparisons vs. Whites; see Table 6).

Respondents also indicated which, if any, among a list of factors would make them feel more comfortable receiving the COVID-19 vaccine. Overall, 61.3% indicated that more testing of the long term effects of the vaccine would make them more comfortable, and this factor did not differ significantly based on race/ethnicity ($\chi^2$ = 2.81, df = 4, n = 1950, ns). A majority of all race/ethnicity groups stated that more information about the vaccine's side effects would make them more comfortable ($\chi^2$ = 8.21, df = 4, n = 1950, ns). About one third (35.4%) of survey participants said that more testing among people of their age group would help increase their comfort levels. Although the overall model of the association between race/ethnicity and this variable was significant ($\chi^2$ = 12.17, df = 4, n = 1950, p < .05), there were no significant differences seen between racial/ethnic groups (see Table 7).

**Table 7. Factors reported to increase comfort receiving the COVID-19 vaccine.**

| | Race/Ethnicity | | | | | |
|---|---|---|---|---|---|---|
| | White | Black | Hispanic | Asian | Other | Total |
| **More testing of long-term effects** | 61.4%$_a$ | 60.4%$_a$ | 62.8%$_a$ | 69.7%$_a$ | 56.2%$_a$ | 61.3% |
| **More information about side effects** | 57.6%$_{a, b}$ | 59.8%$_{a, b}$ | 63.0%$_b$ | 72.7%$_b$ | 51.9%$_a$ | 58.6% |
| **More testing on people of my age** | 34.5%$_a$ | 40.4%$_a$ | 38.8%$_a$ | 42.4%$_a$ | 24.4%$_b$ | 35.4% |
| **More testing on people of my gender** | 15.2%$_a$ | 26.4%$_b$ | 26.3%$_b$ | 21.9%$_{a, b}$ | 13.1%$_a$ | 18.3% |
| **More testing on people of my race/ethnicity** | 11.8%$_a$ | 41.3%$_b$ | 23.7%$_c$ | 28.1%$_{b, c, d}$ | 14.6%$_{a, d}$ | 17.7% |
| **More medical professionals of my race/ethnicity discussing it in the media** | 9.7%$_a$ | 28.1%$_b$ | 23.7%$_b$ | 21.2%$_{b, c}$ | 12.2%$_{a, c}$ | 14.4% |
| **More celebrities or social media influencers of your race/ethnicity discussing it in the media** | 4.3%$_a$ | 9.0%$_b$ | 7.6%$_{b, c}$ | 6.1%$_{a, b, c}$ | 3.1%$_{a, c}$ | 5.3% |
| **None of the above** | 22.6%$_a$ | 16.2%$_{b, c}$ | 14.8%$_{b, c}$ | 6.3%$_c$ | 22.1%$_{a, b}$ | 20.3% |

Notes: Hispanics may be any race, and members of other groups indicated they were non-Hispanic.

Each subscript letter denotes a subset of race/ethnicity categories whose column proportions do not differ significantly from each other at the p < .05 level.

Percentages total more than 100% because survey respondents could select multiple answers.

Among other factors that would make respondents more comfortable about receiving the vaccine, several differences emerged. The overall model of race/ethnicity and desire for more testing on people of one's own race/ethnicity was significant ($\chi^2$ = 130.71, df = 4, n = 1950, p < .05). Black and Hispanic respondents were significantly more likely than Whites to say more testing on individuals of their own race/ethnicity would make them comfortable (41.3% for Blacks, 23.7% for Hispanics and 11.8% for Whites (both comparisons vs. Whites, p < .05; Table 7). Race/ethnicity was also significantly associated with desire for more medical professionals of one's own race/ethnicity and more celebrities or social media influencers discussing the vaccine in the media ($\chi^2$ = 81.00 and $\chi^2$ = 13.01, respectively, both df = 4, n = 1950, p < .05). Black and Hispanic respondents were also significantly more likely than Whites to say more medical professionals of their race/ethnicity and more celebrities of their race/ethnicity discussing the vaccine in the media would make them more comfortable (Table 7). Interestingly, although the gender balance of the racial/ethnic groups did not differ, desire for more testing on people of one's own gender varied significantly by race/ethnicity ($\chi^2$ = 34.01, df = 4, n = 1950, p < .05). Black and Hispanic respondents (26.4% and 26.3%, respectively) were more likely than Whites (15.2%) to say more testing on people of their gender would increase their comfort (both comparisons to Whites p < .05).

## Discussion and conclusions

The research results suggest that Black and Hispanic American individuals are more hesitant than US Whites to receive the COVID-19 vaccine. When asked how long they wanted to wait before receiving the COVID-19 vaccine after the vaccine became broadly available, Blacks and Hispanics were significantly more likely than Whites to indicate that they wanted to wait for over a year and were significantly less likely than Whites to indicate that they wanted to get vaccinated immediately. Blacks and Hispanics were also less likely than Whites to report that they would encourage their family members to get the vaccine. Blacks were more likely to feel that the COVID-19 vaccine was not something they trusted, a result that echoes prior research on reluctance for other vaccines, such as the HPV vaccine [25–27]. This hesitancy towards the vaccine could allow the spread of COVID-19 to persist within ethnic minority communities, which have already been disproportionately impacted by the pandemic.

Possible reasons for this lack of trust and willingness to receive the COVID-19 vaccine may be lack of adequate information or even the spread of misinformation about the vaccine among such communities. Our results indicate that Black respondents were significantly more likely to believe that the vaccine is dangerous, more harmful than getting COVID-19, going to cause people to catch COVID-19, unsafe, and not worth the risk. They were also less likely to believe that the vaccine would be effective. Historical explanations for this lack of trust, such as the Tuskegee experiments, could be one factor, especially given the greater strength of overall findings among Black respondents. Katz et al. [26] suggested that more extensive vaccine information and multimodal patient support may help counteract HPV vaccine hesitancy driven by mistrust; similar approaches should be tested for COVID-19 vaccine hesitancy.

Blacks and Hispanics were significantly more likely to report that it is important for a medical professional of their race/ethnicity to endorse the vaccine before they take it. Fu, Haimo-witz and Thompson [25] previously found that endorsement by race-concordant medical experts was a significant driver of trust in vaccine information for African-American parents. Hernandez et al. [28] identified similar effects among Latina college women considering the HPV vaccine. As there are currently far fewer doctors and healthcare providers of these ethnic groups when compared to White providers, the unwillingness or hesitation of ethnic minorities to readily receive the vaccine may be easier to understand.

Due to these disparities, having medical professionals from Hispanic and Black communities discuss the vaccine may be particularly important. Sandra Lindsay, a Black medical professional, was the first person in the United States to receive the COVID-19 vaccine. She acknowledged the Black community's greater medical mistrust due to historical incidents like the Tuskegee experiments, and cited it as the primary reason she wanted to receive the vaccine publicly: "That was the goal today. . . Not to be the first one to take the vaccine, but to inspire people who look like me, who are skeptical in general about taking vaccines [29]." The second person in the US to receive the vaccine, Dr. Yves Duroseau, was also Black and noted the importance of personally illustrating the vaccine's safety [29]. As the COVID-19 vaccine becomes more widely available to the general public, frequent salient examples of Black and Hispanic medical professionals receiving the vaccine and discussing its benefits could significantly reduce vaccine hesitancy among people of color. Furthermore, Paterson et al. [30] observed that healthcare providers are among the most trusted sources of vaccine information. Given this, healthcare providers' own vaccine status and attitudes are particularly important so that they can accurately inform their patients. However, healthcare worker COVID-19 vaccine hesitancy itself is a potential challenge. Lucia, Kelekar and Afonso [31] found that nearly a quarter of medical students were reluctant to receive the vaccine upon regulatory approval [also see 32, 33]. Paterson et al. [30] noted that making vaccination for healthcare providers convenient and inexpensive, as well as increased training about vaccines and patient communication strategies, can increase vaccine compliance among patients. Given the influential role that Black and Hispanic medical professionals play in encouraging vaccine uptake among Black and Hispanic patients, it is critically important to make it as easy as possible for them to receive the vaccine. US Blacks and Hispanics in this study were significantly more likely than Whites to state that more testing of the vaccine on their racial/ethnic group is needed. Researchers, too, have noted the challenge of achieving adequate representation of Black and Hispanic patients in medical research [e.g., 34, 35]. Our results suggest that the effectiveness and safety of the vaccines among minority study participants should be further highlighted. Promoting the dissemination of accurate research information, including the experiences of minority vaccine recipients, could demonstrate the positive effects of the vaccine in these groups, encouraging them to see the vaccine as safe and effective and increasing vaccination uptake.

Our findings also have more general implications for the use of media to highlight the importance of diverse healthcare spokespeople to help reduce Black and Hispanic vaccine hesitancy. Increased media coverage and representation of Hispanic and Black medical professionals endorsing vaccines and discussing infectious diseases may foster greater trust among Hispanic and Black individuals and encourage them to see vaccines as more trustworthy, safe, and effective. In addition, media could be used to describe past, current, and future vaccine clinical trials and the successful experiences among Black and Hispanic study participants. Our results also suggest that improved, culturally tailored messaging and communication tools to get such messages across could help to improve vaccination rates within minority communities. As the Black American and Hispanic American communities are culturally distinct and each contain a variety of ethnic groups, it is important to develop comprehensive communications that take the diversity of these communities into account. Culturally competent approaches are necessary when developing the most successful methods to increase vaccine acceptance among minority groups. These are important areas for future research.

The current study is not without limitations. The study was conducted at a single point in time, soon after COVID-19 vaccines became widely available. COVID-19 vaccine-related attitudes are likely to continue to evolve as more people receive the vaccine and share their personal experiences within their social groups; this factor alone may result in increased trust and

vaccine uptake. However, the opposite result may also occur: rare incidents of vaccine side effects may be seen as newsworthy and attract attention through media and informal word-of-mouth, worsening vaccine-related fear and hesitation. During the time this survey was conducted, the vaccines that had received Emergency Use Authorization were the mRNA vaccines formulated by Pfizer-Biontech and Moderna. As additional vaccines become widely available vaccine perceptions may change. Additional research may further analyze vaccine attitudes and beliefs as the pandemic evolves and new vaccines are available. In addition, as the pandemic continues, new viral variants and mutations may impact attitudes towards vaccination. Further research should include questions assessing opinions towards vaccination in the context of viral variants, which may be more or less dangerous than the original SARS-CoV-2 virus.

Ideally, the data analysis would have controlled for the level of income and/or education of survey respondents in order to clarify effect drivers. Given the observed racial/ethnic differences in median income, it is reasonable to ask whether any race/ethnicity effects observed in the analysis may actually be driven by these confounding factors. Further examination of these effects is an opportunity for future research. More extensive research is also needed to examine the mechanisms of vaccine hesitancy among people of color and how their vaccine-related concerns can be overcome. An emerging issue regarding COVID-19 vaccination is that of incomplete treatment: since the Pfizer and Moderna vaccines require two doses administered about three to four weeks apart in order to be maximally effective, receiving both doses during the recommended timeframe is of pivotal importance. Culturally-relevant vaccine communications and administration practices are needed to help achieve this, especially among higher-disease-incidence populations such as US Blacks and Hispanics. Future research could be done using mixed methods study, so that nuanced differences between distinct cultural groups can be explored further. Future studies can also further assess whether promoting the spread of accurate study data that shows the positive effects of the vaccine in Black and Hispanic individuals will reduce COVID-19 vaccine hesitancy. Future research could expand on this finding and use experimental methods to test different variations of health communications in communities of color to examine which more effectively diminish vaccine hesitancy.

## Supporting information

**S1 File. COVID-19 vaccine survey.**
(PDF)

**S2 File. COVID-19 vaccine survey data.**
(CSV)

**S1 Data.**
(XLSX)

## Acknowledgments

The authors thank Dr. Ozlem Equils, MD and the MiOra organization for their ongoing support and our reviewers for their thoughtful comments. We also thank our survey respondents for their generosity in sharing their pandemic experiences and perceptions.

## Author Contributions

**Conceptualization:** Katherine Kricorian.

**Data curation:** Karin Turner.

**Formal analysis:** Karin Turner.

**Investigation:** Katherine Kricorian.

**Methodology:** Katherine Kricorian.

**Project administration:** Katherine Kricorian.

**Resources:** Karin Turner.

**Software:** Karin Turner.

**Validation:** Katherine Kricorian.

**Visualization:** Katherine Kricorian.

**Writing – original draft:** Katherine Kricorian.

**Writing – review & editing:** Katherine Kricorian, Karin Turner.

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
