## [Decision Letter · Decision Letter 0]

25 Mar 2021

PONE-D-21-04327

COVID-19 Vaccine Acceptance and Beliefs among Black and Hispanic Americans

PLOS ONE

Dear Dr. Kricorian,

Thank you for submitting your manuscript to PLOS ONE. After careful consideration, we feel that it has merit but does not fully meet PLOS ONE’s publication criteria as it currently stands. Therefore, we invite you to submit a revised version of the manuscript that addresses the points raised during the review process.

ACADEMIC EDITOR: 

All changes suggested by the reviewer must be addressed in order to consider the manuscript for acceptance into PLOS ONE.

We look forward to receiving your revised manuscript.

Kind regards,

Samantha S. Goldfarb, DrPH

Academic Editor

PLOS ONE

Journal Requirements:

1. Please ensure that your manuscript meets PLOS ONE's style requirements, including those for file naming. The PLOS ONE style templates can be found athttps://journals.plos.org/plosone/s/file?id=wjVg/PLOSOne_formatting_sample_main_body.pdf andhttps://journals.plos.org/plosone/s/file?id=ba62/PLOSOne_formatting_sample_title_authors_affiliations.pdf

2. Please include additional information regarding the survey or questionnaire used in the study and ensure that you have provided sufficient details that others could replicate the analyses. For instance, if you developed a questionnaire as part of this study and it is not under a copyright more restrictive than CC-BY, please include a copy, in both the original language and English, as Supporting Information. Moreover, please include more details on how the questionnaire was pre-tested, and whether it was validated.

3. In your Methods section, please provide additional information about the participant recruitment method and the demographic details of your participants.

Additional Editor Comments (if provided):

Thank you for your submission. Please see the reviewer's comments and provide a comprehensive response to each item.

Reviewers' comments:

Reviewer's Responses to Questions

**Comments to the Author**

1. Is the manuscript technically sound, and do the data support the conclusions?

Reviewer #1: Yes

2. Has the statistical analysis been performed appropriately and rigorously? 

Reviewer #1: Yes

3. Have the authors made all data underlying the findings in their manuscript fully available?

Reviewer #1: Yes

4. Is the manuscript presented in an intelligible fashion and written in standard English?

Reviewer #1: Yes

5. Review Comments to the Author

Reviewer #1: The manuscript is fairly well written however, there is no information about participants other than age, race, gender and ethnicity. There are a number of factors that could influence attitudes about the COVID-19 vaccine uptake but were not included in this study. The location of the participants in the US was not included in the manuscript and can also have profound effects on attitudes about the vaccine. Details of the vaccine and variants should be mentioned along hesitancy observed among health care workers. African Americans and Hispanic-Latin-x communities are culturally distinct and will require a culturally competent approach to vaccine hesitancy which should be mention as a part increasing vaccine acceptance in minority communities.

It is my opinion that the manuscript should be considered for publication after my comments are addressed.

6. PLOS authors have the option to publish the peer review history of their article (what does this mean?). If published, this will include your full peer review and any attached files.

Reviewer #1: No

---

## [Author Response · Author response to Decision Letter 0]

26 Apr 2021

Additional information regarding respondent demographic details has been added to the manuscript in a new table, Table 1.

---

## [Decision Letter · Decision Letter 1]

21 Jun 2021

PONE-D-21-04327R1

COVID-19 Vaccine Acceptance and Beliefs among Black and Hispanic Americans

PLOS ONE

Dear Dr. Kricorian,

Thank you for submitting your manuscript to PLOS ONE. After careful consideration, we feel that it has merit but does not fully meet PLOS ONE’s publication criteria as it currently stands. Therefore, we invite you to submit a revised version of the manuscript that addresses the points raised during the review process

Please submit your revised manuscript within 60 days of receipt of this email. If you will need more time than this to complete your revisions, please reply to this message or contact the journal office at plosone@plos.org. Please include the following items when submitting your revised manuscript:

We look forward to receiving your revised manuscript.

Kind regards,

Marlene Camacho-Rivera, ScD, MPH

Academic Editor

PLOS ONE

Journal Requirements:

Reviewers' comments:

Reviewer's Responses to Questions

**Comments to the Author**

1. If the authors have adequately addressed your comments raised in a previous round of review and you feel that this manuscript is now acceptable for publication, you may indicate that here to bypass the “Comments to the Author” section, enter your conflict of interest statement in the “Confidential to Editor” section, and submit your "Accept" recommendation.

Reviewer #1: All comments have been addressed

Reviewer #2: (No Response)

2. Is the manuscript technically sound, and do the data support the conclusions?

Reviewer #1: Yes

Reviewer #2: Partly

3. Has the statistical analysis been performed appropriately and rigorously? 

Reviewer #1: Yes

Reviewer #2: Yes

4. Have the authors made all data underlying the findings in their manuscript fully available?

Reviewer #1: Yes

Reviewer #2: Yes

5. Is the manuscript presented in an intelligible fashion and written in standard English?

Reviewer #1: Yes

Reviewer #2: Yes

6. Review Comments to the Author

Reviewer #1: June 12, 2021

Editor and Chief

PLOS ONE

Manuscript ID: PONE-D-21-04327

Dear Editor,

My comments regarding manuscript PONE-D-21-04327R1 entitled “COVID-19 Vaccine Acceptance and Beliefs among Black and Hispanic Americans” are enclosed.

Thank you for giving me another opportunity to review this revised manuscript submitted to your journal for publication.

Kind regards,

Donald J. Alcendor, Ph.D.

Associate Professor

Meharry Medical College

Center for AIDS Health Disparities Research

& Department of Microbiology and Immunology

& Obstetrics and Gynecology

Hubbard Hospital 5th Floor Rm. 5025

1005 Dr. D.B. Todd Jr. Blvd.

Nashville, TN 37208

Phone: 615-327-6449

Fax: 615-327-6929

Email: dalcendor@mmc.edu

Associate Professor Adjunct

Department of Pathology, Microbiology and Immunology

Vanderbilt University Medical Center

REVIEW

Dear Author,

The manuscript examines COVID-19 vaccine acceptance and beliefs among Black and Hispanic Americans. The paper is part of a broad, representative survey of US adults (N=1950) in order to better understand vaccine beliefs and explore opportunities to increase vaccine acceptance among these groups. The survey results suggested that Black and Hispanic individuals were less willing than Whites to receive the COVID-19 vaccine immediately.

Overall: The manuscript is fairly well written and my comments have been adequately addressed in the revised manuscript. It is my opinion that the manuscript should now be considered for publication.

Main Comments:

1. Because the COVID-19 pandemic is changing daily, the authors should make a statement in the introduction that notes “at the time of this writing…….”

“At the time of this writing, approximately fifty thousand Black Americans and seventy thousand Hispanic Americans have died from the virus.” A reference for this statement would have been nice. This comment has been addressed in the revised manuscript.

2. The available COVID-19 vaccines that have undergone Emergency Use Authorization should be included in the introduction because this information is changing daily and the updated information should be reflected in the manuscript. At the time of the survey performed for this study there were only 2 EUA approved vaccines, now there are 3, the J&J vaccine was approved February 27, 2021.

This comment has been addressed in the revised manuscript.

3. The term “achieve broad-scale immunity” should be replaced by “herd immunity” which is define at this time to be between (75%-85%) of the US population.

This comment has been addressed in the revised manuscript.

4. The authors should mention the long-standing safety of vaccines given to children and adolescents to attend public school and the extremely low rate of adverse events in this population that represent 24% of the US population. Even more, we have to vaccinate enough of the global population to end this pandemic and we have to curtail the evolution of viral variants that could lead to this virus becoming endemic across the world not just in the US.

This comment has been addressed in the revised manuscript.

5. There is also not one sentence about viral variants and their contribution to the pandemic.

This comment has been addressed in the revised manuscript.

6. Vaccine hesitancy among healthcare providers should be mentioned in the manuscript.

This comment has been addressed in the revised manuscript.

7. There is no information about participants other than age, race, gender, and ethnicity. There is no information on their socioeconomic status, if they have access to the internet, low vs high income, education level, political affiliation, and do they have insurance etc. There many factors that influence attitudes about this vaccine.

This comment has been addressed in the revised manuscript.

8. The other category in Table 1 should be briefly defined.

This comment has been addressed in the revised manuscript.

9. Relating these findings to participant education, income levels, political affiliation etc could have profound effects on the data.

This comment has been addressed in the revised manuscript.

10. The participants location in the US (rural vs suburban, South vs West) could also be mentioned and would likely have a profound effect on the results of this study.

This comment has been addressed in the revised manuscript.

11. African Americans and Hispanic-Latin-x communities are culturally distinct and will require a culturally competent approach which should be mention as a part increasing vaccine acceptance and reducing vaccine hesitancy in minority communities.

This comment has been addressed in the revised manuscript.

Reviewer #2: This revised manuscript seeks to establish the prevalence of racial and/or ethnic differences in COVID-19 vaccine acceptance. Unfortunately, I am being asked to review the resubmission without having been a reviewer of the initial submission. To that end I have a few concerns about the papr that were not previously raised, while with one exception (noted below) the authors seemed responsive to the previous reviewers’ comments.

Primary Concerns that should be addressed:

My main concern is that H4 essentially arises from nowhere, with the background of the paper focusing exclusively on why we would expect racial/ethnic differences, and provides essentially no motivation for this hypothesis, or even why this particular question is an additional focus of the paper. There is ample existing research to draw on to provide support for this hypothesis, but as written this is missing from the current manuscript.

In the text, the authors suggest that care was taken to compare the sample to US composition, and re-weight to adjust for that representation, but no details are provided for those comparisons (e.g., in Table 1) or weighting approaches.

Lesser Concerns for the authors to consider:

A previous reviewer asked how the observed associations are patterned by indicators of SES. The authors claim that such multivariate analyses are not feasible with these data. While the structure may not allow for formal mediation analyses, they seem more than sufficient to include one (or more) of these as controls in a multivariate model. That is, the paragraph on p. 19 is an unconvincing response to this previous reviewer request. I’m not convinced this paper needs that additional layer; but it’s not infeasible as the authors suggest.

The phrase “end the pandemic” (p. 1) is not especially meaningful. The likely outcome is not that SARS-COV-2 will fade away, but that it will become endemic in ways that are manageable at a population level. In other words, this common phrase often brings to mind an unlikely eventuality, and isn’t very helpful here.

The range of estimates are wide for herd immunity. It’s unclear why the authors draw only on high-end estimates, suggesting the need for near universal vaccine coverage. If anything, this focus is likely to undermine the seeming aim of the paper to increase vaccine uptake.

The responses to the timing question (Table 3) could be substantially affected by perceived timing of availability / access to the vaccine. In other words I worry that responses here could essentially be double barreled in that they may capture both how long someone perceives it will be before the vaccines is available to them, and how long they would wait after that point, while it’s only being interpreted to represent the latter. Is there any reason to not be concerned about this? If not, a caveat should probably be made about its interpretation.

The text on p. 12 and the table on p. 13 don’t match. The text suggests the comparisons being made include those reporting both extremely and very important, while it’s clear they only include the “extremely” row from the table.

Admittedly pedantic writing details:

- “Data” are plural. So “CDC data also indicate” (not “indicates”) on p. 2.

- Methodology is the study of methods. A study’s design and analytic strategies are its “Methods” (p. 4).

- “…respondents were less likely to be…” (not “like) on p. 6.

7. PLOS authors have the option to publish the peer review history of their article (what does this mean?). If published, this will include your full peer review and any attached files.

Reviewer #1: **Yes: **Donald J. Alcendor

Reviewer #2: No

---

## [Author Response · Author response to Decision Letter 1]

8 Jul 2021

Dear Reviewer:

Thank you for your thorough review of our manuscript, COVID-19 Vaccine Acceptance and Beliefs among Black and Hispanic Americans. We appreciate your thoughtful comments and are grateful for the opportunity to sharpen our work based on your feedback. Please see our responses to your comments below.

Primary Concerns that should be addressed:

My main concern is that H4 essentially arises from nowhere, with the background of the paper focusing exclusively on why we would expect racial/ethnic differences, and provides essentially no motivation for this hypothesis, or even why this particular question is an additional focus of the paper. There is ample existing research to draw on to provide support for this hypothesis, but as written this is missing from the current manuscript.

We have deleted H4 as a hypothesis and now report the results from those survey questions as exploratory findings. Our intent in including these questions in the survey is to provide potential directions for developing communication plans for addressing the other hypothesized findings. 

In the text, the authors suggest that care was taken to compare the sample to US composition, and re-weight to adjust for that representation, but no details are provided for those comparisons (e.g., in Table 1) or weighting approaches.

We have added the following on page 5 to provide more detail on the data weighting: 

Population values were developed using data from the 2019 US Census American Community Survey. Population percentages were calculated for the matrixed percentage values for age range, gender, race/ethnicity, income range and Census region, and were compared to the same matrixed percentage values within the survey sample. Survey responses within each Age range * Gender * Race/ethnicity * income range * Census region cell were upweighted or downweighted so that the resulting percentages, shown in table 1, reflected the population distribution present in the ACS results. Most cell weights differed negligibly from 1.0.

Lesser Concerns for the authors to consider:

A previous reviewer asked how the observed associations are patterned by indicators of SES. The authors claim that such multivariate analyses are not feasible with these data. While the structure may not allow for formal mediation analyses, they seem more than sufficient to include one (or more) of these as controls in a multivariate model. That is, the paragraph on p. 19 is an unconvincing response to this previous reviewer request. I’m not convinced this paper needs that additional layer; but it’s not infeasible as the authors suggest.

We have revised this section on page 21 of the tracked changes manuscript as follows:

Ideally, the data analysis would have controlled for the level of income and/or education of survey respondents in order to clarify effect drivers. Given the observed racial/ethnic differences in median income, it is reasonable to ask whether any race/ethnicity effects observed in the analysis may actually be driven by these confounding factors. Further examination of these effects is an opportunity for future research.

The phrase “end the pandemic” (p. 1) is not especially meaningful. The likely outcome is not that SARS-COV-2 will fade away, but that it will become endemic in ways that are manageable at a population level. In other words, this common phrase often brings to mind an unlikely eventuality, and isn’t very helpful here.

We have deleted this phrase.

The range of estimates are wide for herd immunity. It’s unclear why the authors draw only on high-end estimates, suggesting the need for near universal vaccine coverage. If anything, this focus is likely to undermine the seeming aim of the paper to increase vaccine uptake.

We have added the following on page 2 of the tracked changes manuscript:

However, Dong et al. hypothesized that vaccination of 60% of the population may be sufficient. Randolph and Barriero noted that herd immunity thresholds can vary widely within populations and are non-uniformly distributed, implying that a single standard for herd immunity may be an oversimplification. 

The responses to the timing question (Table 3) could be substantially affected by perceived timing of availability / access to the vaccine. In other words I worry that responses here could essentially be double barreled in that they may capture both how long someone perceives it will be before the vaccines is available to them, and how long they would wait after that point, while it’s only being interpreted to represent the latter. Is there any reason to not be concerned about this? If not, a caveat should probably be made about its interpretation.

The specific wording of the survey question was “After the COVID-19 vaccine is broadly available, how long do you plan to wait to get the vaccine?” However, the Table 3 heading implies that the survey question wording was the potentially double barreled “How long do you plan to wait to get the COVID-19 vaccine?” We have corrected the table heading and text references on page 10 to clarify this more nuanced question wording. 

The text on p. 12 and the table on p. 13 don’t match. The text suggests the comparisons being made include those reporting both extremely and very important, while it’s clear they only include the “extremely” row from the table.

We have corrected this. 

- “Data” are plural. So “CDC data also indicate” (not “indicates”) on p. 2.

- Methodology is the study of methods. A study’s design and analytic strategies are its “Methods” (p. 4).

- “…respondents were less likely to be…” (not “like) on p. 6.

We have corrected these errors.

Thank you again for your thoughtful review of this manuscript and for providing such constructive feedback. We appreciate your help.

---

## [Decision Letter · Decision Letter 2]

30 Jul 2021

COVID-19 Vaccine Acceptance and Beliefs among Black and Hispanic Americans

PONE-D-21-04327R2

Dear Dr. Kricorian,

We’re pleased to inform you that your manuscript has been judged scientifically suitable for publication and will be formally accepted for publication once it meets all outstanding technical requirements.

Kind regards,

Marlene Camacho-Rivera, ScD, MPH

Academic Editor

PLOS ONE

Additional Editor Comments (optional):

Reviewers' comments:

Reviewer's Responses to Questions

**Comments to the Author**

1. If the authors have adequately addressed your comments raised in a previous round of review and you feel that this manuscript is now acceptable for publication, you may indicate that here to bypass the “Comments to the Author” section, enter your conflict of interest statement in the “Confidential to Editor” section, and submit your "Accept" recommendation.

Reviewer #2: All comments have been addressed

2. Is the manuscript technically sound, and do the data support the conclusions?

Reviewer #2: Yes

3. Has the statistical analysis been performed appropriately and rigorously? 

Reviewer #2: Yes

4. Have the authors made all data underlying the findings in their manuscript fully available?

Reviewer #2: Yes

5. Is the manuscript presented in an intelligible fashion and written in standard English?

Reviewer #2: Yes

6. Review Comments to the Author

Reviewer #2: (No Response)

7. PLOS authors have the option to publish the peer review history of their article (what does this mean?). If published, this will include your full peer review and any attached files.

Reviewer #2: No

---

## [Editor Report · Acceptance letter]

9 Aug 2021

PONE-D-21-04327R2 

COVID-19 Vaccine Acceptance and Beliefs among Black and Hispanic Americans 

Dear Dr. Kricorian:

I'm pleased to inform you that your manuscript has been deemed suitable for publication in PLOS ONE. Congratulations! Your manuscript is now with our production department. 

Kind regards, 

on behalf of

Dr. Marlene Camacho-Rivera 

Academic Editor

PLOS ONE